# Suboptimal Chest Radiography and Artificial Intelligence: The Problem and the Solution

**DOI:** 10.3390/diagnostics13030412

**Published:** 2023-01-23

**Authors:** Giridhar Dasegowda, Mannudeep K. Kalra, Alain S. Abi-Ghanem, Chiara D. Arru, Monica Bernardo, Luca Saba, Doris Segota, Zhale Tabrizi, Sanjaya Viswamitra, Parisa Kaviani, Lina Karout, Keith J. Dreyer

**Affiliations:** 1Department of Radiology, Massachusetts General Hospital and Harvard Medical School, Boston, MA 02114, USA; 2Mass General Brigham Data Science Office (DSO), Boston, MA 02114, USA; 3Department of Diagnostic Radiology, American University of Beirut Medical Center, Beirut 11-0236, Lebanon; 4Department of Radiology, Azienda Ospedaliera G. Brotzu, 09134 Cagliari, Italy; 5Department of Radiology, Hospital Miguel Soeiro—UNIMED, Sorocaba 18052-210, Brazil; 6Department of Radiology, Pontificia University Catholic of São Paulo, São Paulo 05014-901, Brazil; 7Department of Radiology, Azienda Ospedaliera Universitaria di Cagliari, 09123 Cagliari, Italy; 8Medical Physics and Radiation Protection Department, Clinical Hospital Centre Rijeka, 51000 Rijeka, Croatia; 9Radiology Department, Iran University of Medical Sciences, Tehran 14535, Iran; 10Department of Radiodiagnosis, Sri Sathya Sai Institute of Higher Medical Sciences, Whitefield 560066, India

**Keywords:** artificial intelligence, chest X-ray, computer-assisted image processing, quality improvement, radiography

## Abstract

Chest radiographs (CXR) are the most performed imaging tests and rank high among the radiographic exams with suboptimal quality and high rejection rates. Suboptimal CXRs can cause delays in patient care and pitfalls in radiographic interpretation, given their ubiquitous use in the diagnosis and management of acute and chronic ailments. Suboptimal CXRs can also compound and lead to high inter-radiologist variations in CXR interpretation. While advances in radiography with transitions to computerized and digital radiography have reduced the prevalence of suboptimal exams, the problem persists. Advances in machine learning and artificial intelligence (AI), particularly in the radiographic acquisition, triage, and interpretation of CXRs, could offer a plausible solution for suboptimal CXRs. We review the literature on suboptimal CXRs and the potential use of AI to help reduce the prevalence of suboptimal CXRs.

## 1. Introduction

The best introduction to suboptimal chest radiographs (CXRs) and artificial intelligence (AI) might start with the words of famous American composer Duke Ellington (1899–1974), “a problem is a chance for you to do your best”. In the context of suboptimal CXRs, the words imply a dire need for the best solutions, including education and AI. At the same time, a growing body of evidence urges a cautionary approach to AI and reminds us of the words of the legendary World War II correspondent Edward R. Murrow (1908–1965), “Our major obligation is not to mistake slogans for solutions”.

While several AI-related studies report promising use-case scenarios for AI applications in CXRs, users must recognize the limitations of AI as well. Prior studies have reported on AI applications in triaging, segmentation, detection, and diagnosis of radiographic findings, as well as risk stratification, outcome prediction, and image optimization of CXRs [1,2,3,4]. Conversely, others draw attention to flaws in research and commercial CXR-AI models with regulatory clearance from the United States Food and Drug Administration (FDA) [5].

In this article, we discuss issues, causes, impact, and potential solutions related to suboptimal CXRs; a similar approach can apply to radiographs of other body parts as well as to other imaging modalities. We review the literature on suboptimal CXRs and the potential use of AI to decrease their prevalence.

## 2. Optimal and Suboptimal CXRs: The Criteria

The American College of Radiology (ACR)-Society of Pediatric Radiology (SPR)-Society of Thoracic Radiologist (STR) Practice Parameters for the performance of chest radiography and European guidelines on quality criteria for diagnostic radiographic images provide guidelines for the specifications of the exam [6,7]. These guidance documents define optimal CXRs as those with optimal exposure as visibility of the lung parenchyma at a mid-gray level; inclusion of both lung apices and costophrenic angles; optimal position without overlapping of scapulae and arms on the lungs; centering of the vertebral column between the clavicles; appropriate definition of lower thoracic vertebrae and retrocardiac pulmonary vessels; and collimation to limit exposure to body parts beyond thorax. 

Figure 1 illustrates various causes of suboptimal CXRs as a result of deviations from optimal radiography techniques. Suboptimal CXRs can be related to low or high gray-level exposure of lung fields (related to under- or over-exposure); non-inclusion of entire lungs from apices to costophrenic angles, rotation or oblique acquisition without centering of the vertebral column between the clavicles, chin, arm, or removable foreign bodies (such as lockets, zippers, coin, and watches) obscuring parts of anatomy. Other deficiencies include inadequate definition of lower thoracic vertebrae and retrocardiac pulmonary vessels, low lung volumes from poor inspiratory breath-hold, technical inadequacy resulting in increased noise and processing and cassette-related artifacts, lack of proper collimation to limit exposure beyond lungs, unintended lordotic or angulated projections.

## 3. Suboptimal CXRs: The Problem

Poor quality exams not only affect the diagnostic interpretation but also have an economic impact. The national average cost of CXRs in the US is $420, with substantial variations across different locations and sometimes in the same region [8]. Given that 40% of the 3.6 billion worldwide imaging studies performed every year are CXRs, the cost implication of rejected and repeated suboptimal CXRs can be enormous [9]. A repeat radiograph is associated with increased radiation exposure, additional time and resources, workflow issues, diagnostic delays, and potential limitations and pitfalls in interpretation with persistent suboptimality.

Issues related to suboptimal radiography do not have simple solutions, as in the quote, “A problem well-stated is a problem half-solved”, from John Dewey (1859–1952), an American philosopher, psychologist, and educational reformer. Although suboptimal radiography is often related to errors in its acquisition, not all causes leading up to suboptimal CXRs stem from inadequate technologist training or a lack of attention to detail. Often, and especially for portable CXRs in acutely sick patients on life support or severely debilitating conditions, there is little a technologist or high-end acquisition technologies can do to obtain optimal CXRs. However, in a world besotted with technological innovations, where solutions often search for problems or amplify some issues to emerge as saviors, it is critical to clearly define the magnitude of the problem from suboptimal CXRs before justifying conventional mitigating steps or proposing cutting-edge remedies with AI.

With an ever-increasing use of imaging [10], there is a need for improved quality control. Quality control in radiography is vital for all three main types of radiography, including conventional/film radiography, computed radiography (CR), and digital radiography (DR). Conventional/film radiography has several limitations, including dose reduction, fixed non-linear grey-scale response, incompatibility with the PACS (Picture Archiving and Communication System), and environmental and storage issues [11]. Although CR is less expensive than DR and offers multiple-size detector cassettes, it can produce poor-quality radiographs and is labor-intensive. Overexposure leading to suboptimal or rejected conventional radiographs can be missed with CR and DR due to the ability to correct the window level and width on the viewing workstation.

DR results in higher quality radiographs and opportunities to enhance or manipulate radiographs after acquisition to reduce exposure-related issues. However, image manipulation with DR cannot fix problems outside of radiographic exposure, such as patient positioning, low lung volumes, obscuring body parts or artifacts, clipped anatomy of the lungs, and inadequate collimation. Thus, all radiography technologies are vulnerable to suboptimal quality and, therefore, require surveillance and quality control measures.

Various causes of suboptimal CXRs and reject rates are summarized in Table 1. A 2015 study from Tschauner et al. reported that only 4% of CXRs fulfilled all criteria for optimal pediatric CXRs [12]. The study evaluated the quality of pediatric radiographs for meeting the European guidelines with the primary focus on optimal collimation of CXRs since the optimal field size is vital in reducing radiation dose. The authors reported only 49% of radiographs were performed at the peak of inspiration and 76% of examinations without rotation or tilting. From a review of 80-0 CXRs, Okeji et al. [13] subcategorized CXR quality based on patient details, anatomical markers, anatomic coverage, full inspiration, artifacts, position of scapula, radiographic exposure, blurring, rotation, and darkroom processing faults. Only 17% of CXRs met the optimal quality criteria, with inadequate collimation being the most common cause of suboptimal CXRs (83%, *n* = 664/800 CXRs) [14].

Several publications have reported on the reject rate for radiographs [12,13,14,15,16,17,18,19,20,21]. Reject rate refers to suboptimal radiographs that are rejected or discarded, and often require repeat radiographs to obtain a diagnostic quality radiograph. For CXRs, prior research reported the reject rates varying between 4% and 15% (Table 2) [14,15,16,17,18,19,20,21]. Jabbari et al. evaluated 5695 radiographs in Iran and reported an 11% repeat or reject rate. Problems related to exposure (over- and under-exposure) were the commonest cause of rejection. Other causes of suboptimality included position faults, patient motion, and processing faults leading to artifacts or exposure-related issues. The pelvis and upper limb radiographs had the highest and lowest repeat rate of 14% and 4%, respectively [14]. A similar study from Namibia reported errors in patient positioning as the major cause of rejection, followed by issues related to under or over-exposure [15]. The overall departmental reject rate was 8%, with a 10% reject rate for CXRs. The 16% (mammogram), 13% (skull), 10% (cervical spine), and 8.3% (thoracic spine) repeat rates were higher than the overall average [15]. 

Foos et al. performed a study in a university and community hospital setting to analyze the reject rate for CR examinations [16]. CXRs were the most frequently performed examinations and had a reject rate of 9% and 8.8% at the university and community hospitals, respectively. The reason for rejection presented in their study included clipped anatomy, positioning errors, patient motion, artifacts, clipped markers, incorrect markers, and low and high exposure index. Shoulder, hip, and spine radiographs had a reject rate of 9–11%, 10%, and 8–11%, respectively [16].

Jones et al. reviewed 66,063 radiographs from one year using an automated recording system [17]. Default reasons for technologists to select when rejecting radiographs were positioning issues, wrong patient identification number, exposure errors, test images, and artifacts. A blank field was also provided for technologists to enter a free text cause for rejecting radiographs if necessary. A total of 6002 radiographs were rejected over the duration of the study from multiple modalities. They reported a reject rate of 3% for portable and 30% for decubitus CXRs. The reject rates for pelvis, shoulder, humerus, cervical, thoracic, and lumbar spine radiographs were 19%, 14%, 13%, 12–25%, 11–27%, and 10–16%, respectively. Positioning errors accounted for 77% of the rejection, while 10% of rejected radiographs were from exposure-related issues. Their study also highlights the importance of an automated radiation exposure system to address the issues related to the reject rate [17].

The study by Sadiq et al. conducted a reject repeat analysis of a plain radiograph in Nigeria [18]. The 37% reject rate for CXRs was significantly greater than the overall reject rate of 29%. Under- and over-exposure accounted for 36% and 24% of rejections, while clipped anatomy, excessive patient rotation, and artifacts contributed to 22%, 5%, and 4% rejections, respectively. Compared to the CXRs, postnasal space, paranasal sinus, and pelvis radiographs had higher reject rates with 58%, 43%, and 67%, respectively.

Ali et al. conducted a study during the COVID-19 pandemic to evaluate the reject rate [20]. They reported an overall reject rate of 17%. The causes of rejection in the order of its frequency were positioning, artifacts, motion, collimation, labeling, exposure errors, and machine/detector faults. The 23% reject rate for CXRs was 15% higher than the overall average. In their study, skull radiographs (45%) had the highest reject rate, followed by pelvis (35%), abdomen (28%), and neck (21%).

These studies highlight the prevalence and causes of issues related to the quality of radiographs. The high variation between the studies on reject rates could be related to the subjective rejection of radiographs by the technologists at different sites.

## 4. Suboptimal CXRs: Impact and Issues

The substantial difference between what is deemed as suboptimal (as high as 83–96% in some studies) [12,13] versus the 4–15% reject rate [14,15,16,17,18,19,20,21] is likely related to the fact that suboptimal CXRs are far more common, and therefore, less often rejected. A low reject rate might also imply that the CXRs either have minor reasons (such as CXRs with a minor degree of rotation) or unsolvable issues (such as low lung volumes in ventilated or debilitated patients). Alternatively, radiology services might have a high degree of tolerance for suboptimal CXRs, resulting in fewer rejected and reacquired images for reasons related to costs, workflow, and ability to resolve underlying etiologies of suboptimal CXRs. The ongoing COVID-19 pandemic has led to an increase in the rejection rate in our department from <5% to as high as 9% in our quaternary healthcare practice due to a combination of staff shortages juxtaposed with increased demand for CXRs while maintaining a safe distance and minimizing patient contact.

The impact of suboptimal CXRs is non-trivial. For example, in a critically ill or unstable patient, clipped lung apices or overlying anatomy can limit the evaluation of pneumothorax, apical pneumonia, or lesions. Likewise, an underexposed image or one with low lung volumes can limit the evaluation of lung bases and the position of lines and tubes. Excessive patient rotation can affect the evaluation of lung, hila, and cardiomediastinal abnormalities. Artifacts can mimic lesions, triggering additional diagnostic tests or repeat radiographs and causing patient anxiety. Suboptimal radiographs can also lead to misinterpretation resulting from false positive or false negative interpretations of CXR findings.

Beyond the adverse impact of suboptimal CXR on diagnostic interpretation, reacquisition can delay patient care, which is especially important for urgent or critical findings. They can negatively affect patient workflow and cause patient inconvenience, especially in outpatient settings where patient recall might be necessary to repeat CXRs. The latter can happen with film/conventional radiography and CR, where images are not available for immediate viewing. With DR, technologists have immediate access to radiographs and can verify optimality and reacquire before letting the patient leave. Reacquisition also increases the technologists’ workload. Given the profound clinical importance of CXRs, a lower frequency of suboptimal CXRs is desirable but a challenging goal. Yet, once a problem is stated, perhaps perseverance can bring success with the words of Amelia Earhart (1897–1937), the first female aviator to fly solo across the Atlantic—“The most difficult thing is the decision to act, the rest is merely tenacity”.

Despite extensive guidelines on CXR image quality [6,7], mitigation of suboptimality and reject rates remains problematic. Quality control and improvement are challenging. For one, they are labor-intensive, time-consuming, and often require manual review of radiographs. Although DR is conducive to immediate mitigation with rejection and reacquisition, additional radiographs still entail additional radiation exposure to patients. With a quick image quality review and rejection analysis with DR, there are minimal delays and workflow issues, but such an option is tedious for conventional radiography and CR. Another benefit of the DR system pertains to post-acquisition image enhancement and manipulation to salvage some suboptimal radiographs.

A focus on efficiency and productivity often requires technologists to maximize patient throughput and give the backseat to quality control measures. Usually, the rejected radiographs are not archived and only become statistics for monitoring databases and information. While these statistics are valuable tools for audit and surveillance, they represent a lost opportunity to “show and tell” or “see and remember”. These should be considered opportunities for what to avoid and improve among the causes of recurrent problems resulting in suboptimal or rejected radiographs. Therefore, retention of rejected radiographs in some form is a valuable educational resource for preventing the recurrence of some suboptimal radiographs [22].

Beyond documentation of a radiation event, suboptimal and rejected radiographs can provide information on the cause and need for rejection and, more importantly, whether the repeat radiograph mitigated the issue with rejected radiographs. As the technologists acquiring the radiographs are usually tagged to the image, they can receive personalized feedback on the errors while avoiding punitive actions [23]. Individual technologists must not be held accountable for the poor quality, as the cause is usually multifactorial. For example, it may not be possible to avoid underexposed suboptimal CXRs in morbidly obese patients or CXRs with clipped lung bases in severely hyperinflated lungs. Positive reinforcement with rewards can motivate and inspire radiographers to put in additional effort and attention to optimizing radiographic acquisition.

It is essential to record the reject rates for both portable and fixed radiography equipment. In addition, the conventional mitigation strategy of auditing and review is necessary to understand the scope and impact of suboptimal CXRs. Coupled with continuous learning and feedback on quality issues with radiography, these can help mitigate suboptimal radiography but require additional staffing on quality assurance personnel such as in our institution.

## 5. Mitigation: The New Direction

While surveillance and education can reduce suboptimality and reject rates, mitigation might benefit from new thinking given the multifactorial causes of suboptimality. Perhaps Albert Einstein (1879–1955) was correct when he stated, “We cannot solve our problems with the same level of thinking that created them”. So, is AI the new level of thinking or mitigation for suboptimal CXRs and other radiographic examinations?

AI is ushering in a new revolution in medicine, and medical imaging is at the forefront of AI applications due to its massive digital footprint. For example, in radiography, and specifically for CXRs, several AI algorithms triage and detect radiographic findings [1,2,3,4,24]. There is little doubt that some causes of suboptimality are not always solvable, such as exposure issues in an extremely large patient or low lung volumes in critically ill, unconscious patients. For others, such as clipped anatomy or overlapping structures or artifacts, AI can help.

AI algorithms from some commercial entities (such as Qure.ai, Annalise, and Carestream) also target qualitative aspects of CXRs. For example, fixed Carestream’s radiography units use AI before and after image acquisition. For positioning the patient, they utilize two RGBD cameras (Red, Green, Blue, and Depth) to collect patient information and transfer it to an AI-based pose-detection algorithm and classifier. The information on fixed DR units helps automatically adjust the Bucky height to the patient and helps radiographers. This smart positioning system communicates essential aspects of ideal CXR to the radiographers, such as patient contact with Bucky, center alignment, patient orientation, tilt, and hand position. Such information can help radiographers avoid patient positioning errors [25]. Another AI-based feature (Smart Noise Cancellation, Carestream) based on a deep convolutional neural network trained to predict input image noise can result in 2 to 4× noise reduction without loss of sharpness of anatomical structures [26]. With noise cancellation, users can reduce radiation dose by up to two-fold, especially relevant for neonates and small children.

AI algorithms can also determine the patient size and adjust or adapt automatic exposure control settings on some fixed radiography units to ensure adequate quality. Another use of cameras and AI on radiography units involves the recognition of shoulder joints to determine the correct collimation field size and settings. While reducing radiation dose to body regions beyond the chest, this AI-based smart collimation feature saves the radiographer’s time and decreases subjectivity with manual collimation adjustment [25]. Furthermore, post-acquisition noise reduction filters can help improve the quality of CXRs, as reported in several studies [27,28]. Fukui et al. reported the potential for up to 72% radiation dose reduction for portable DR with the use of noise reduction software to improve the image quality of low radiation dose CXRs [27].

Many radiography vendors (such as AGFA, Fujifilm, GE, and Siemens) also offer options such as auto-positioning for CXRs using AI integration and cameras. For example, Siemens YSIO X.pree X-ray system deploys an AI-integrated 3D camera for automatic body-part detection and collimation adjustment in less than 0.5 s [29].

Two AI vendors have introduced algorithms that analyze some causes of suboptimal CXRs. Annalise AI algorithm [30] for CXRs evaluates patient rotation, cervical flexion, underinflation (low lung volumes), under- or over-exposure, and clipped or obscured anatomy. A similar AI algorithm from Qure.ai assesses incompletely imaged CXRs and specifies the excluded anatomy (such as left lung, left apex, or left costophrenic angle), patient rotation, under-or over-exposure, and incomplete inspiration (low lung volumes) [31]. 

We have developed a suite of home-grown AI algorithms to assess different causes of suboptimal CXRs on the COGNEX Vision Pro Deep Learning platform, which allows non-programmers to build AI models without having any programming knowledge [32]. Our models identify clipped anatomy (such as apices and lung bases), over- and under-exposed CXRs, patient rotation, obscured anatomy by chin or arms projecting on the chest, and low lung volume due to inadequate inspiration, as shown in Figure 2. We intend to deploy these AI algorithms to perform a post-acquisition image quality audit of CXRs. Such audits will help track suboptimality and develop case-based continuous learning for radiographers.

The trajectory of AI applications in CXRs suggests an ongoing and expanding suite of AI applications using camera-mounted, AI-enabled radiography units to automate positioning, centering, rotational, and collimation tasks. Such systems can help reduce errors relative to legacy radiography units. In addition, post-acquisition, AI algorithms can evaluate CXRs and prompt radiographers to repeat radiographs as needed. Although best integrated into the radiography units, such image quality assessment AI algorithms can help conduct retrospective audits for suboptimal CXRs and identify the scope and magnitude of suboptimal CXRs.

While AI algorithms can help avoid and identify causes of suboptimal CXRs, radiographers’ participation is critical. At the time of preparing this manuscript, there were no publications on AI use in suboptimal CXRs. In addition, several questions remain unanswered on the accuracy and performance of these AI applications and algorithms, such as on portable CXRs and in the presence of complex patient anatomy. We hope that our review will trigger further research and verify the robustness and generalizability of available AI solutions. 

The ultimate question for the future is whether the trajectory of scientific developments beside AI will bring completely autonomous robotic radiographic units to remove human errors in radiography. While such development might reduce human errors, portable radiography, particularly in an acutely sick patient, is challenging beyond manual and technical issues. Such challenges are likely to continue due to issues of complex patient geometry, anthropometry, and sometimes from an expanding and advanced life support system that continues to advance in parallel to the science that helps mitigate the existing issues.

## 6. Conclusion

In summary, a substantial proportion of CXRs are suboptimal and require reacquisition. However, the reacquisition of rejected CXRs involves additional radiation exposure, workflow issues, and delays in patient care. While awareness, audit, and continuous education represent vital strategies to mitigate the high frequency of suboptimal CXRs, automation with AI-integrated cameras and enabled algorithms will likely help the quality of CXRs.

## Figures and Tables

**Figure 1 diagnostics-13-00412-f001:**
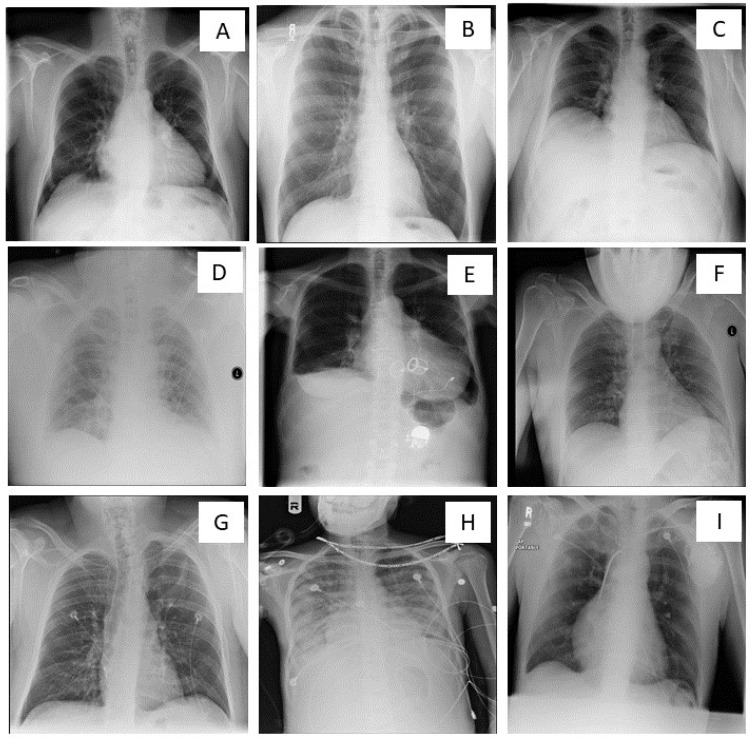
Optimal and suboptimal chest X-rays. (**A**)—Optimal quality chest X-ray. Suboptimal chest X-rays (**B**–**I**) resulting from (**B**)—non-inclusion of lung apices and costophrenic angles; (**C**)—low lung volume/ inadequate inspiration; (**D**)—under-exposure; (**E**)—over-exposure; (**F**)—chin overlying the lung fields; (**G**)—patient rotation; (**H**)—foreign body(necklace) overlying the lung field; (**I**)—artifact in the lower part of the image obscuring part of left costophrenic angle.

**Figure 2 diagnostics-13-00412-f002:**
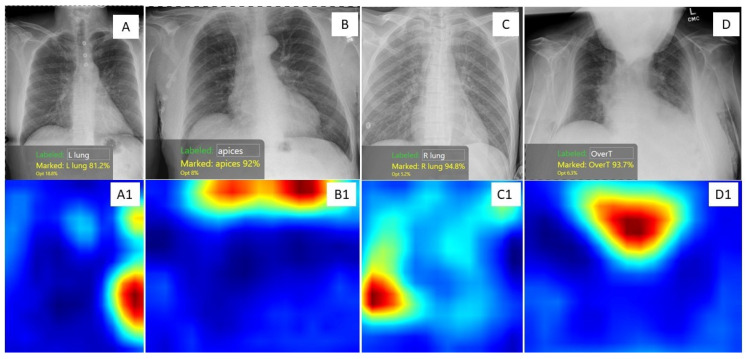
CXRs and corresponding heat maps of four adult patients demonstrating various causes of suboptimal CXRs. The heat maps were produced by four of our AI models built on the Cognex Platform. (**A**) CXR with clipped left costophrenic angle was identified as red areas on heat map image **A1**; (**B**) Suboptimal CXR with non-inclusion of lung apices was marked in red color on heat map image **B1**; (**C**) Suboptimal CXR with clipped right costophrenic angle was identified as a red area on heat map image **C1**; (**D**) Suboptimal CXR due to patient’s chin obscuring lung and mediastinum was identified as a red region on heat map image **D1**.

**Table 1 diagnostics-13-00412-t001:** Summary of prior publications on various causes of suboptimal CXRs. (Key: DR—digital radiography; CR—computed radiography; FR—film/conventional radiography; %—percentage).

Publications	Radiography	Causes	% Poor Quality/Reject Rate
Suboptimal CXR causes			
Tschauner S et al. (2015) [12]	DR	Inspiration Rotation	51% 24%
Okeji MC et al. (2017) [13]	DR & CR	Collimation Scapula in the lung field Darkroom processing faults Poor exposure Rotation	83% 38% 34% 28% 27%
Rejected CXR causes			
Jabbari N et al. (2011) [14]	FR	Under-exposure Over-exposure Position fault Patient motion Processing fault Others	2.77% 3.63% 1.37% 0.93% 1.16% 0.98%
Benza C et al. (2018) [15]	CR	Over-exposure Under-exposure Double-exposure Gridlines Anatomical marker Artifacts positioning	0.34% 1.36% 0.34% 0.23% 0.57% 0.45% 6.46%

**Table 2 diagnostics-13-00412-t002:** Tabular summary of reject rates for CXRs in prior publications. (Key: DR—digital radiography; CR—computed radiography; Conventional—conventional/film radiography; FR—film/conventional radiography).

Reject Analysis	Country	Radiography	Total CXRs	Rejected CXRs	Reject Rate
Foos et al. (2009) [16]	USA	CR	102,678	5134	5%
Jabbari et al. (2011) [14]	Iran	FR	5695	626	11%
Jones et al. (2011) [17]	USA	CR & DR	27,409	1096	4%
Sadiq et al. (2017) [18]	Nigeria	FR	4171	1557	37%
Benza et al. (2018) [15]	Namibia	CR	882	88	10%
Atkinson et al. (2019) [19]	Australia	DR	39,185	2743	7%
Ali et al. (2021) [20]	Pakistan	DR	3858	579	15%
Arbese et al. (2021) [21]	Ethiopia	FR	1690	152	9%

## Data Availability

No new data were created or analyzed in this study. Data sharing is not applicable to this article.

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
