# Peer review of "Suboptimal Chest Radiography and Artificial Intelligence: The Problem and the Solution"

_diagnostics, 2023, doi:10.3390/diagnostics13030412_

Round 1

Reviewer 1 Report

The context is quite redundant. Minor revision is suggested.

Author Response

Dear Reviewer, 

Thank you for the review and suggestions. We have made changes and improved the manuscript to reflect the importance of the context.

Reviewer 2 Report

This paper provides a comprehensive review of recent works on suboptimal CXRs and the potential use of AI to mitigate suboptimal CXRs, which is important for urgent or critical findings. The authors point out the problems, summary the solutions, and give their thoughts about the future direction. To summarize, this paper is well-organized and easy to follow. My only concern is that the authors should proofread their paper and avoid some typos.

This paper is well organized and easy to follow. I agree to accept the current version.

Author Response

Dear Reviewer, 

Thank you for the kind note and suggestions. We apologize for the errors in the manuscript. We have made minor changes and corrected the typos and mistakes. 

Reviewer 3 Report

A substantial proportion of CXRs are suboptimal and require reacquisition. However, the reacquisition of rejected CXRs involves additional radiation exposure, 

workflow issues, and delays in patient care. While awareness, audit, and continuous education represent vital strategies to mitigate the high frequency of suboptimal CXRs, automation with AI-integrated cameras and enabled algorithms will likely help the quality of CXRs.

1. The contribution is not stated clearly.

2. The abstract is not clear : it must not contain results which are not presented and substantiated in the main text and should not exaggerate the main conclusions.

3. In the introduction, what key theoretical perspectives and empirical findings in the main literature have already informed the problem formulation? What major, unaddressed puzzle, controversy, or paradox does this research address?

4. Why does it need to be addressed?  Why it should be now and past? Challenges and solution is not clearly written. it would be great to give table.

5. Further, in the introduction, what is the recent knowledge gap of the main literature that the author needs to write this research? What we have known and what we have not known? What is missing from current works? Please explain and give examples!

6. In terms of the knowledge gap, it will be best if the research challenge/knowledge gap could be stated in one article or more articles in the main literature. Assure that you have included all key articles in the main literature. Mention all of these article. https://doi.org/10.1007/s12553-022-00700-8  https://doi.org/10.3390/diagnostics12102549  https://doi.org/10.1016/j.compbiomed.2022.106083 https://doi.org/10.3390/s20041068   https://doi.org/10.1007/978-3-030-04061-1_17 How does explainability helps need to mentioned.

7 Research question must be explicitly stated in the introduction. Show how the main literature informs the formulation of your research question(s).

8 Besides, it feels to me you just put the tables and figures in result section rather than their explanation. Try to explain each table and figure in detail in the result section. Discussion is a separate section.

9 In terms of theoretical contribution, show the theoretical novelty that your solution offers. How does this novelty distinguish your work from other similar works?

10 The requirements of a practical implications is not clearly stated, how do the findings help the health organizations? Please explain and give examples! Assure that any recommendation is clear and actionable for organizations.

11 As to practical implications, show how your recommendation is timely. As to both theoretical contributions and practical implications, show how your research potentially influences health organizations and individuals to think, behave, or perform through multifaceted forms and channels

Author Response

Dear Reviewer,

Thank you for the review. Concerns are addressed in the letter to the editor.

Reviewer 4 Report

1. The manuscript is well focused on advanced research topic on chest radiography combined with AI technique.

2. The problems and solutions are well presented in the paper.

3. Organization of the paper is good.

4. Introduction of the paper could have few more information (which doesnt affect my decision).

5. Study images presented and tables clearly highlight on the significance of proposed work.

6. Conclusion is clear and the objective of the study is met out.

7. References are fine.

Author Response

Dear Reviewer, 

Thank you for the kind note and suggestions. We have provided more information related to the problem of the suboptimal Chest radiographs in separate headings 2 and 3. Therefore the introduction is concise in the manuscript. We have made minor changes to the manuscript. Thank you again for reviewing the manuscript.

Reviewer 5 Report

Reviewer comments

Although CXR plays an important role on the diagnosis and management of diseases, a substantial proportion of CXRs are suboptimal and require reacquisition, leading to delays in patient care and pitfalls in radiographic interpretation. Therefore, it is likely to employ the AI-integrated cameras and enabled algorithms to mitigate the suboptimality and improve the quality of CXRs.

The manuscript is well-written and well-organized. The objective is well-articulated and reached. The figures and tables are presented in a clear and appropriate manner and are consistent with the description in the text. The results and analysis presented in the manuscript are interesting for this field and Diagnostics is the appropriate journal to submit it. But there are still some points that the authors should consider, as described in the following. Also, some suggestions are provided, in case the authors consider them interesting to carry out.

When using abbreviations in academic writing, the first time you mention a phrase that can be abbreviated, spell it out in full and provide the abbreviation in parentheses. Use only the abbreviation thereafter. The authors should check the entire manuscript and see if abbreviations are used properly. We can see some abbreviations are defined in the text, but the full expression is still used later.

In Figure 1, it would be helpful to add an optimal CXR as a control image.

Figure 2 is not mentioned in the main text of the manuscript.

In line 144, “Table 1” appears twice.

In Table 1 and Table 2, it is better to use square brackets [] to mention the reference numbers (e.g., [12], [13], [14]), rather than parentheses ().

In Table 1 and Table 2, please adjust the width of column and make a phrase in one line. For example, “Darkroom processing faults” is shown in two lines in Table 1, resulting in difficulty to find the corresponding reject rate. “Radiography” is shown in two lines in Table 2.

In line 159-160, “Other such as mammogram, skull, cervical and thoracic spine radiographs had 16%, 13%, 10 and 8.3% repeat rate, which were higher than the overall average.” 10 should be 10%. Does the “repeat rate” mean “reject rate”?

In line 162-163, “DR- digital. radiography”. The dot is not necessary between “digital” and “radiography”.

In line 196, “Reject rate of CXRs was 15% lower than the overall average”. Does it mean the reject rate of CXRs was 2% since the overall reject rate is 17%? Or we should understand this sentence as “Reject rate of CXRs was 15% , which was lower than the overall average”.

In line 202-203, “4. Suboptimal CXRs: Impact and Issues” should be the heading in this section and should be written in one line.  

In line 330, the supplementary material is not included in the PDF version of the manuscript.

Author Response

Dear Reviewer, 

Thank you for the valuable suggestions and feedback. We have addressed all the comments and made changes to the manuscript to reflect the same. Kindly find the response to the comments below:

  1. In Figure 1, it would be helpful to add an optimal CXR as a control image: Optimal CXR has been added to figure 1.
  2. Figure 2 is not mentioned in the main text of the manuscript: Fig 2 has been mentioned in the body of the main body of the manuscript.
  3. In line 144, “Table 1” appears twice: changes have been made.
  4. In Table 1 and Table 2, it is better to use square brackets [] to mention the reference numbers (e.g., [12], [13], [14]), rather than parentheses (): parenthesis has been changed to the square bracket.
  5. In Table 1 and Table 2, please adjust the width of column and make a phrase in one line. For example, “Darkroom processing faults” is shown in two lines in Table 1, resulting in difficulty to find the corresponding reject rate. “Radiography” is shown in two lines in Table 2: table width has been adjusted to fit into a single line.
  6. In line 159-160, “Other such as mammogram, skull, cervical and thoracic spine radiographs had 16%, 13%, 10 and 8.3% repeat rate, which were higher than the overall average.” 10 should be 10%. Does the “repeat rate” mean “reject rate”? : Yes, reject and repeat rate are synonymous, and the sentence has been changed to 10%.
  7. In line 162-163, “DR- digital. radiography”. The dot is not necessary between “digital” and “radiography”: dot has been removed.
  8. In line 196, “Reject rate of CXRs was 15% lower than the overall average”. Does it mean the reject rate of CXRs was 2% since the overall reject rate is 17%? Or we should understand this sentence as “Reject rate of CXRs was 15% , which was lower than the overall average”: reject rate was 23%, so the sentence has been changed to "The reject rate of CXRs was 23% which was 15% higher than the overall average. "
  9. In line 202-203, “4. Suboptimal CXRs: Impact and Issues” should be the heading in this section and should be written in one line.: changes have been made.
  10. In line 330, the supplementary material is not included in the PDF version of the manuscript: the supplementary material is figure 2. Therefore, the sentence has been changed.

Round 2

Reviewer 3 Report

I have not received point to point response from the authors.

Author Response

(The authors gave the same response as above.)
